# The Relevance of Thomas Berry for 21st Century Catholicity

**Cristina Vanin**

Religious Studies, St. Jerome's University, Waterloo, ON N2L3G3, Canada; cristina.vanin@uwaterloo.ca

**Abstract:** The ecological crisis continues to be identified as the most significant social breakdown in the world. One of the important foundational influences on the development of an adequate religious response is the thought of cultural historian Thomas Berry. He affirmed the critical role that the world's religions have in developing a spirituality that supports the sacrifices, visions, and dreams needed to live in an integral way with the Earth's community of life. Such a spirituality provides the psychic energies we need to adequately respond to the crisis. The author of this article argues that Berry's thoughts continue to be relevant, especially in the context of the emergence of a renewed sense of Catholicity. This article presents an overview of the breadth and depth of the study that led to Berry's articulation of a new human orientation needed to reverse the path of devastation. It offers Berry's insights into the reasons why it is difficult for Christianity to effectively respond to the present crisis and calls for a new Catholicity that functions out of the comprehensive context of an evolutionary and emergent universe.

**Keywords:** ecological crisis; religion; spirituality

## 1. Introduction

The publication of John Paul II's 1990 World Day of Peace message, *Peace with God, Peace with all Creation*, is regarded as the time when the leadership of the Roman Catholic church started to directly speak about ecological concerns: "In our day, there is a growing awareness that world peace is threatened not only by the arms race, regional conflicts and continued injustices among peoples and nations, but also by a lack of due respect for nature, by the plundering of natural resources and by a progressive decline in the quality of life . . . " (John Paul II 1990) At the core of the ecological crisis is the fact that "[i]t is manifestly unjust that a privileged few should continue to accumulate excess goods, squandering available resources, while masses of people are living in the conditions of misery at the very lowest level of subsistence" (John Paul II 1990). In other words, if Catholicism is to adequately respond to ecological issues, the principles of Catholic social teaching need to be extended to include the Earth.

The ecological awareness that emerged in the 1980s saw connections between the suffering that human beings are experiencing and the damage that is being caused to the planet. A healthy planet starts to be understood as another condition for fulfillment and the achievement of the common good, not just for human persons but also for all other-than-human beings.

This development of a Catholic response led to the publication of the 2015 encyclical, *Laudato Si': On Care for Our Common Home* (Francis 2015). As with previous writings on social issues, Pope Francis combined the riches of the church's teaching with the findings of experts in a variety of fields as he reflected on this critical problem of our times. What is new and significant here is that Pope Francis said that concern and care for the Earth is not optional; instead, it is an integral part of the church's teaching on social justice.

One of the most important influences on the development of this Catholic response is the thought of cultural historian Thomas Berry, who advocated for a transformation of human relationship to the natural world starting in the 1960s. Berry's attention to ecological concerns emerged after many years spent studying and teaching about the world's eastern,

western, and indigenous religious and cultural traditions. While Berry's prophetic voice is no longer alone in its calls for recognizing our interconnectedness with the natural world, it continues to provide one of the richest and most expansive articulations of the root causes of our current ecological crisis and what is needed in an adequate response. In the context of this Special Issue on 'Catholicity' as the emergent process of 'faith making wholes,' and participative faith functioning within a world characterized by evolution (Delio 2015; John 2011), this article argues for the relevance of Berry's thought by presenting the background to his ecological insights, the particular challenges faced by Christianity in responding to the ecological crisis, and the implications of his ideas regarding a new story of the universe for such a renewed understanding of Catholicity, especially within the Catholic church.

## 2. The Emergence of Thomas Berry's Ecological Concerns

Thomas Berry's comprehensive vision of the universe story emerged after he spent several decades studying eastern, western, and indigenous religious and cultural traditions. With William Theodore de Bary, he founded the Asian thought and religion seminar at Columbia University. He taught Asian religions at Seton Hall University (1956–61) and St. John's University (1961–65). He then went to Fordham University, where he taught from 1966 until 1979 and developed a graduate program on the history of religions. He established the Riverdale Center for Religious Research in New York City, where the ten thousand books in the library included the Latin texts of the Christian church fathers, the Sanskrit texts of Hinduism and Buddhism, and the Chinese classics of Confucianism and Taoism. Berry's approach to these religions was to attend to not only their history but also their spiritual dynamics and their significance for the contemporary world.[1]

Berry's study of the thought of Pierre Teilhard de Chardin influenced his growing appreciation of time as developmental and his persistent statements that we now know the universe to be an unfolding reality, what Berry calls 'cosmogenesis'—an ever-evolving rather than static cosmos. Teilhard also provided Berry with the idea that the universe has both a physical and a psychic character. This implies that if there is human consciousness and if humans have evolved from the Earth, then some kind of consciousness has been present in the process of evolution from the beginning. In other words, matter is not dead or inert; rather, it is a numinous reality, a reality with both physical and spiritual dimensions. Furthermore, there are various forms of consciousness; in humans, consciousness is reflective. This means that humans have a special role in the evolutionary process as a highly complex species with the capacity for critical reflection. It is because of Teilhard that Berry continuously talks about the evolution of the universe as the most comprehensive context for understanding who human beings are in relation to other forms of life and the universe itself. Human beings have a particular responsibility for the way in which the evolutionary process moves forward on Earth.

Through his extensive study of cultural traditions, Berry developed a deep appreciation for indigenous traditions. He regularly talked about the ways in which such native traditions have a particular relationship to place, recognizing the sacredness of land, seasons, and all creatures such as animals, birds, and fish. He appreciated that their respect for creation stems from their honoring of the Creator and from their reverence for life as a gift. For Berry, indigenous wisdom will be critical as human beings meet the ecological demands of our time.

Following his early experiences in childhood; his life experiences of worldwide war, depression, recovery, and growth; and his education, Berry's ideas about a 'New Story' began to emerge in the 1970s (Berry 1978, 1988). This historian of culture was thinking deeply about the magnitude of the devastation that was taking place on the planet. The change being caused by such devastation is unparalleled because the fundamental systems of life are threatened. Berry argued that all human institutions—political, economic, educational, and religious—have been contributing to the establishment and reinforcement of a "radical discontinuity between the human and other ways of being" (Berry 1999). He

saw that human beings had come to think of themselves—consciously or not—as deeply alienated from the Earth. This is a consciousness that has given all rights to humans and that regards other-than-human beings as having reality and value only when used by humans. When this kind of consciousness is operating, we find it easy to exploit what we regard as completely other. We learn to regard and treat certain human beings, such as the poor, indigenous peoples, people of color, and people with disabilities as less than those of us in positions of privilege.[2] This alienation, along with the West's commitment to increased industrialization and commercialization, has spread across the entire planet.

In his book, *The Great Work: Our Way into the Future*, Berry stated that, while we do not choose the time in which we are born, or our culture, or the particular moment of history in which we find ourselves, "the nobility of our lives . . . depends upon the manner in which we come to understand and fulfill our assigned role" (Berry 1999). That assigned role in our time is the difficult transition from being a deleterious human presence on the Earth to becoming a mutually-enhancing presence in the world—what Berry referred to as the Ecozoic Era. While Berry argued that all human institutions need to undertake this work, as primarily a historian of religious and cultural traditions, Berry gave extensive attention to the role that the world's religions have in bringing the Ecozoic Era into being. He noted that, "[s]o far Christians have not distinguished themselves by their concern for the destiny of Earth. Now, however, this care has become the special role, not only of Christians, but of all humankind, a role no other age could fulfill, a role so important that there may not *be* another truly human age in the future if the present conflict of human with Earth is not resolved" (Berry 2014). Given this essay's concern with Berry's relevance to a 21st century Catholicity, I focus on Berry's understanding of Christianity's particular contributions to ongoing human alienation from the natural world.

## 3. Berry's Articulation of the Limitations of Christian Responses

Berry traced this alienation and consequent loss of intimacy with the natural world initially back to the early meeting of Christianity with Greek humanism. One of the consequences of this encounter was the formation of a strong anthropocentrism that eventually led to the exaltation of the human over the natural world, and a loss of any sense that the divine is revealed in the natural world. He specifically looked at the cultural and religious responses to the plague that struck Europe in the 14th century and argued that people were so surprised by the extent of the loss of human life that a negative attitude toward nature arose. He found evidence of these responses in writers such as Jacopo Passavanti, a Dominican preacher from the 1350s, who "talked about the pains of hell and the need for a severe spiritual regime to negate the body and the world so as to turn toward the heavenly realm," and Thomas á Kempis, who proposed "a spiritual orientation based on detachment from the concerns of everyday life" in his work, *The Imitation of Christ* (Berry 2009b). Berry suggested that, by the 15th century, this kind of devotionalism and spirituality had developed to such a point that humans were completely separated from any meaningful interest or concern in this world. In his words, "[i]nstead of delight and pervasive experience of the divine in the world's beauty, wonder, and awesomeness, there developed a psychic–spiritual withdrawal from too intimate relations with one's surroundings" (Berry 2009a). Christians had come to regard the Earth as dangerously seductive and as causing our human alienation from the divine.

In addition, in response to this experience of extreme loss of human life during the plague, Christianity developed a focus on a redemptive spirituality that spoke about being freed and saved from this menacing Earth into a blissful heavenly realm. "The salvific, redemptive traditions of the West tend to save humans out of the temporal order or to assign meaning to the temporal order in terms of a 'salvation history,' with an ultimate goal outside of time" (Berry 2009a). Aided by the emphasis that was placed on the human soul and the loss of any sense of soul in the natural world, Christians no longer saw the natural world as an essential aspect to their spiritual salvation; they could acknowledge that degradation of the natural world is unfortunate but not spiritually significant. For



Berry, "[t]his attitude creates an immense psychological barrier to our Christian intimacy with Earth. We are here, as it were, on trial, to live amid the things of this world but in thorough detachment from them . . . That we truly belong to this world is difficult for us" (Berry 2009a).

Unfortunately, at least since the rise of modern science, Berry argued that human beings have lost any awareness of the natural world communicating a sense of the divine. Instead, "[t]he universe has become a collection of objects, not a communion of subjects" (Berry 2009a, 2009b). The world has become an 'it' that we can use as we see fit, rather than something we can relate to as a 'thou,' as subjects to subjects. We no longer truly relate to and hear the voices of the rivers, the mountains, the seas, the trees, and the animals as intimate ways in which the spirit of the divine is present to us. Berry frequently spoke about the deep alienation that human beings live in relation to the universe as a deep cultural pathology. We live in and with the natural world and yet are so separate from it that we do not even recognize how alienated we truly are. The irony is that we have more knowledge of the universe than any people in history has ever had, but it is not the type of knowledge that leads us to have an intimate presence to other creatures and that makes it possible to live in a world that we consider to be truly meaningful in and of itself. As Berry himself put it:

> While we have more scientific knowledge of the universe than any people ever had, it is not the type of knowledge that leads to an intimate presence within a meaningful universe . . . Our world of human meaning is no longer coordinated with the meaning of our surroundings. We have disengaged from that profound interaction with our environment that is inherent in our nature (Berry 1999).

Although such intimacy is still experienced by some, it is no longer as extensive as it once was. We can think about the culture of consumerism in which each of us lives, a culture that has shifted ancient holy days into holiday shopping days. By contrast, earlier cultures drew meaning and value from the natural rhythms of the cosmos and the cycles of nature. For example, early Christian liturgies corresponded to the movements of the sun. Feasts were celebrated in relationship to the seasons so that, for example, it is no accident that in the Northern Hemisphere, we celebrate the feast of Christmas around the time of the winter solstice and the feast of Easter in the heart of spring. Now, however, with the loss of cosmological meaning, our social order is governed not by the rhythms of the natural world but by the rhythms of commerce, industry, and consumption.

Berry linked the negative attitude that humans have towards the natural world to another common element in religious traditions, namely the idea of the need for rebirth. "In antiquity, nothing was undertaken in the human order by humans alone. It had to be done in alliance with both cosmic and spiritual processes. Any integral activity involved a three-fold aspect: human, spiritual, and natural" (Berry 2009a). However, this ancient sense of rebirth as bringing us into a higher, sacred, spiritual realm—as involving a spiritual experience of divine presence—has been replaced. With the emergence of modernity, attention shifted to the experience of a new humanity, a new rationality, and a new social order emerging in human history. This historical vision in the West has led humanity to trying to achieve rebirth on our own, through our human endeavors and through our scientific and technological mastery of the natural world, with little connection to the psychic and spiritual dimensions of the natural world. For Berry, this 'infrahistorical drive' and its connection to a millennial vision when all of life would be transformed and the human condition finally healed led to an inherent drive to build the world that we had always envisioned. However, the result of this vision and drive is that, "[i]gnorant of any spiritual significance in what we are doing, we remain profoundly dissatisfied, inwardly starved, spiritually and humanly debilitated, and unable to carry out successfully our finest endeavors" (Berry 2009b). Berry suggested that we are not conscious of the cost to our souls and our sense of ourselves, of being deprived of an intimate relationship with the Earth and the universe and of the loss of cosmological meaning. Consequently, Berry

sought revelatory experiences and spiritualities that would motivate Christians to heal the human relationship to the Earth.

## 4. Implications of Berry's Thought for 21st Century Catholicity

### 4.1. A New Cosmological Context

Berry's perspective was that classical civilizations and indigenous traditions were better equipped for this critical kind of spiritual transformation because they lived within a comprehensive world, one in which the human, the divine, and the cosmic were present to each other in an intimate way. In other words, human endeavors were understood within a cosmological or universe context. Unfortunately, this is not the case in our contemporary world. We do not truly live in a universe; rather, we live in cities and countries, in economic systems, in cultural and, perhaps, religious traditions. As Berry said, "[e]veryone lives in a universe; but seldom do we have any real sense of living in a world of sunshine by day and under the stars at night. Seldom do we listen to the wind or feel the refreshing rain except as inconveniences to escape from as quickly as possible" (Berry 1999). We regard the natural world as simply a backdrop to our human undertakings. Indeed, we are not even aware that we need to have an integral and intimate relationship with the natural world around us.

Since the mid-20th century, Berry argued that a new revelatory experience is available to us as human beings through the scientific understanding of the universe as time-developmental or 'cosmogenesis'—a universe that moves through evolutionary processes and transformations. It is this new story of the emergence of the universe that can provide humans with a new orientation and perspective, as well as a new context for connection, purpose, meaning, and action. For Berry, 'the new story' is the comprehensive basis that will nurture an intimate relationship between humans and the natural world. As a functional cosmology, Berry was convinced that this story can help us to understand how things came to be as they are, where we are now, and how our future can be meaningful.[3]

If we are going to recover a world of meaning and adequately respond to the ecological crisis, then this understanding of the universe must become the comprehensive foundation for all of our cultural and religious traditions. However, it cannot only be understood in materialistic terms because such a limited understanding cannot provide the motivation and sensitivities needed to deal with the ecological crisis. In our time, we need to know that the story of the universe provided to us by science is a sacred story.

Berry regarded the story of the universe as a sacred story because it is the most significant way in which the divine is being revealed to us today. It is also our primary story, our primary community, our primary religious reality—though it is not primary in the sense of first in the order of time nor in the sense of first in importance or value. For Berry, the universe is our primary sacred reality in the sense of most basic, or most fundamental: that which is meant to first awaken us to the divine. This experience of the divine "has given us a new sense of the universe, a new sense of the planet earth, a new sense of life, of the human, even a new sense of being Christian . . . The story has its imprint everywhere, and that is why it is so important to know the story. If you do not know the story, in a certain sense you do not know yourself; you do not know anything" (Berry 1991).

### 4.2. The Challenge of Relating to the Earth as Sacred

Berry was convinced and hopeful that today's religious traditions have the capacity to contribute to the arduous and liberating journey to help humanity move into a larger, more comprehensive, and deeply spiritual realm of being. This is because "[o]ur sense of who we are and what our role is must begin where the universe begins. Not only does our physical shaping and our spiritual perception begin with the origin of the universe, so too does the formation of every being in the universe" (Berry 1999). Everything that exists in the universe is genetically related to everything else. Community is at the heart of the nature of existence. As Berry said, "There is literally one family, one bonding, in the

universe, because everything is descended from the same source ... On the planet earth. ... we are literally born as a community; the trees, the birds, and all living creatures are bonded together in a single community of life" (Berry 1991). This contention is related to Berry's insistence that we need to think of the universe as a community of subjects rather than a collection of objects. A truly intimate relationship with the natural world overcomes our experience of being so locked up in ourselves as humans that we cannot be present to other beings.

An ongoing problem for the contemporary Catholic church is that it has not yet fully accepted this scientific understanding of the universe and is unable to see its religious value. It continues to be faithful and focused on the past, such that it is isolated from the larger community of life on the planet in the present. It has maintained an intense focus on Jesus Christ, concerned itself only with our interior relationship with the divine, and encouraged Catholic Christians to focus on the gospels, carry out works of mercy, and follow the discipline of the Christian life. In and of themselves, none of these invite adequate concern for the planet or the universe, nor is there yet any real sense that the planet and universe are relevant to our spiritual lives.

Another problem is that Christianity has been overly concerned with the transcendence of the divine over the created order and with the transcendence of the human over the natural world. Attention to divine transcendence has meant a corresponding loss of understanding that there is a primary and inherent relationship between the human and the divine within the natural world itself. Berry noted that the Christian affirmation of incarnation requires that we develop a greater sense that humans are an integral part of the Earth community: "If God has desired to become a member of [the Earth] community, humans themselves should be willing to accept their status as members of that same Earth community" (Berry 2009a). The Catholic church needs to accept and integrate the insights of contemporary science into all aspects of the tradition if it is to have relevance because "the human community and the natural world will go into the future as a single sacred community or neither will survive in any acceptable manner" (Berry 2009a).

### 4.3. Areas for a Renewed Understanding of Catholicity

Berry indicated how remarkable it is that Christianity goes beyond the notion of the divine as pure simplicity, a notion that is common to other religions, to speak of the inner life of the divine as community. He offered the three-fold model of differentiation, inner articulation, and communion as a way to talk about the Trinity in the context of our present understanding of the universe. Not surprisingly given his thinking that the universe reflects the divine, Berry identified these as the three basic principles that characterize the universe. The first principle of differentiation is the basic direction of the evolutionary process. Not only is each individual being different from every other being in the universe, as the first principle points out, but each individual being also has its own unique spontaneities, which Berry called 'the sacred depth of the individual.' "Each being in its subjective depths carries that numinous mystery whence the universe emerges into being" (Berry 1999). The principle of communion speaks to the fact that there is a unity to the divine and to the universe, a way in which the presence of each individual is felt throughout the whole of the universe. For Berry, this capacity for bonding makes it possible for such a vast variety of beings to come into existence and to exist in unity with one another.

Berry argued that "[w]e need a spirituality that emerges out of a reality deeper than ourselves, a spirituality that is as deep as the Earth process itself, a spirituality that is born out of the solar system and even out of the heavens beyond the solar system" (Berry 2009b). For such an adequate spirituality to continue to emerge, we will need to articulate the new story as fully as possible. This is precisely wherein the world's religious traditions have their role because "[o]nly religious forces can move human consciousness at the depth needed. Only religious forces can sustain the effort that will be required over the long period of time during which adjustment must be made. Only religion can measure the magnitude of what we are about" (Berry 2009a).[4]

One example of the emergence of a cosmological and Earth spirituality within the Catholic church is the Passionist religious community of Canada (part of the larger Passionist community of which Thomas Berry was a member), which, in the 1970s, began to integrate Passionist spirituality with the story of the universe at the Holy Cross Centre for Ecology and Spirituality. One of the ways in which the center incorporated this more comprehensive spirituality was through building a pilgrimage walk on the property called 'Stations of the Cosmic Earth.'[5] The ancient spiritual practice of a pilgrimage takes one into the new and comprehensive story of the university, both the joys and beauty of its ongoing emergence and evolution, and the increasing suffering of the Earth community. This walk responds to Berry's call for humans "to establish rituals for celebrating these transformation moments that have enabled the universe and the planet Earth to develop," (Berry 2006) so that we would come to appreciate the sacred character of the universe. The opportunity to walk these stations is an opportunity to overcome our alienation and nurture a relationship of profound presence and intimacy to the comprehensive community of life.

Another example is the Ignatius Jesuit Centre in Guelph, Ontario, which has not only developed its own set of 'Stations of the Cosmos' but also developed a number of eight day ecology retreats rooted in the discipline of the Spiritual Exercise of St. Ignatius of Loyola. The four 'weeks' of the Exercises are structured around the life, death, and resurrection of Jesus Christ. In the ecology retreats, this rich, extensively practiced spiritual discipline is transformed into the context of the comprehensive sacred universe. These retreats help people to develop the practice of deep and intentional attention to not only the words of sacred scripture but also the natural world in order to learn the sacredness of the Earth community and to develop the capacity to live out of the cosmological horizon of divine loving.[6]

It is the religious and indigenous traditions that can help us appreciate that the story has a dimension that transcends the physical—that, as Teilhard suggested, the universe has been a psychic and spiritual reality as well as a physical reality from the start. These traditions can help us to understand that our human story is absolutely integral with the story of the universe. When this happens, "[t]hen we can see that this story of the universe is in a special manner our sacred story, a story that reveals the divine particularly to ourselves, in our times; it is the singular story that illumines every aspect of our lives—our religious and spiritual lives as well as our economic and imaginative lives" (Berry 2009b). A new Catholicity can help us to recover a capacity for being in communion with the Earth and understanding ourselves as integral with the universe process.

In Berry's thinking, we should consider this period of Earth history as a moment of grace, that is, another privileged moment when great transformations can occur. This 'cosmological moment of grace' points to the fact that, for the first time, human beings have a new experience and understanding of the deepest mysteries of the universe and the planet. This moment is also a historical and religious moment of grace precisely because understanding the scientific story of the universe as a sacred story helps us to truly know our place in the emerging and evolving universe. The sacred story of the universe could help us to know where we came from and what our responsibility is for the future of the Earth (Berry 1999).

The recognition of ourselves as integral members of the Earth community makes it possible for us to experience a real communion among subjects rather than objects. We begin to be able to hear the voices of all creatures. We learn of the integral relationship among all of the members of the community and that nothing is what it is without everything else. "[W]e form a single sacred society with every other member of the Earth community, with the mountains and rivers, valleys and grasslands, and with all the creatures that move over the land or fly through the heavens or swim through the sea" (Berry 2009b). Every component member of this community of life has its own identify, dignity, and inner spontaneity—its sacred dimension. Even more significantly for a new Catholicity, it is the

community of the whole universe that is the intention of the divine in the experiences of creation, redemption, and transformation.

## 5. Conclusions

When Thomas Berry first began to put together the elements of his ideas regarding the universe story, after years of pondering and brooding over the plight of the planet and after years of studying the great religions of East and West, there were almost no religious voices speaking with him. While there are now many more voices addressing the ecological crisis, Berry's invitation and challenge to think about who we are as human beings, how our human lives are related to the life of the rest of creation, what might be the meaning and purpose of all of life, and how we should live in the world continue to be particularly poignant and profoundly needed.

In *The Great Work*, Berry talked about his experience of growing up in North Carolina in the early part of the twentieth century and the impact of that experience on the fundamental orientation of his life. He recalled a time when he was about eleven years old, and his family moved to the edge of town to a new house that was situated on an incline that led to a little creek; across the creek was a meadow.

> It was an early afternoon in late May when I first wandered down the incline, crossed the creek, and looked out over the scene. The field was covered with white lilies rising above the thick grass. A magic moment, this experience gave to my life something that seems to explain my thinking at a more profound level than almost any other experience I can remember . . . .As the years pass this moment returns to me, and whenever I think about my basic life attitude and the whole trend of my mind and the causes to which I have given my efforts, I seem to come back to this moment and the impact it has had on my feeling for what is real and worthwhile in life . . . .Whatever preserves and enhances this meadow in the natural cycles of its transformation is good; whatever opposes this meadow or negates it is not good. My life orientation is that simple. It is also that pervasive (Berry 1999).

Berry was deeply concerned that human beings, especially children, no longer have these kinds of experiences that help them to know the magnificence of life as celebration; consequently, we lack an attitude of gratitude as our response toward the gift of life. He lamented what has happened to human children: "For children to live only in contact with concrete and steel and wires and machines and computers and plastics, to seldom experience any primordial reality or even to see the stars at night, is a soul deprivation that diminishes the deepest of their human experiences" (Berry 1999). Or again, "[o]ur children no longer learn how to read the great Book of Nature from their own direct experience or how to interact creatively with the seasonal transformations of the planet. They seldom learn where their water comes from or where it goes" (Berry 1999). He argued that we need to provide our children with experiences and opportunities that will help them to develop a deep intimacy and profound presence with the natural world. As he made clear in the dedication to *The Great Work*, when Berry spoke of the children, he had in mind all the children of the Earth community, "the children who swim beneath the waves of the sea, to those who live in the soils of the Earth, to the children of the flowers in the meadows and the trees in the forest, to all those children who roam over the land and the winged ones who fly with the winds, to the human children too, that all the children may go together into the future in the diversity of their regional communities".[7]

In the ongoing emergence of a renewed Catholicity for the 21st century, the thought of Thomas Berry continues to challenge the contemporary Catholic church to awaken and understand ourselves as part of an amazing and sacred universe story. This profound shift is not simply a matter of deepening our appreciation for the natural world, especially for its beauty. Instead, Berry's thought continues to invite all religions, including the Catholic church, to enter deeply into the dynamics of creation and the story of the universe, to nurture a new depth of intimacy with the natural world, and to integrate this intimacy

and relationship into all aspects of the thought and life of the Catholic church. Such a cosmological orientation and spirituality make it possible for us to become, with God, knowers, co-healers, and lovers of all of creation.

**Funding:** This research received no external funding.

**Institutional Review Board Statement:** Not applicable.

**Informed Consent Statement:** Not applicable.

**Data Availability Statement:** Not applicable.

**Conflicts of Interest:** The author declares no conflict of interest.

## Notes

1   For a fuller presentation of Thomas Berry's history, see (Tucker 2006; Berry 2014; Heather 2014).

2   See *Laudato Si'* for Pope Francis' discussion of the need to overcome this culture of indifference to the suffering of the poor and the Earth with a culture of encounter and dialogue.

3   For a discussion of the need to integrate cosmological meaning with anthropological meaning, see (Vanin 2019). See also (Ormerod and Vanin 2016).

4   See the recent documents from the Canadian Catholic Bishops (Canadian Conference of Catholic Bishops 2001, 2004, 2007) as examples of ways in which a new Catholicity that nurtures an intimate relationship with the natural world is emerging within the Catholic church. See also Donal Dorr's (2012) integration of an option for the Earth into Catholic social teaching and the anthology, *Living Cosmology*, which explores the implications of Berry's cosmology as that has been presented in 'Journey of the Universe.' (Mary and Grim 2016).

5   The Maryknoll Ecological Sanctuary in the Philippines have also developed a meditative walk on the universe story that integrates indigenous crafts and images. The Ignatius Jesuit Centre of Guelph, Ontario, opened a similar set of 25 stations called Stations of the Cosmos. Both of these centers were inspired by Thomas Berry's work.

6   See Louis M. Savary, *The New Spiritual Exercises,* for a further example of how the Spiritual Exercises of St. Ignatius can evolve into a comprehensive universe context (Savary 2010).

7   See Pope Francis' encyclical, *Laudato Si',* at the heart of which is the question: "What kind of world do we want to leave for those who come after us, to children who are now growing up?"

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
