# Peer review of "The Relevance of Thomas Berry for 21st Century Catholicity"

_religions, doi:10.3390/rel12100878_

Round 1

Reviewer 1 Report

This article represents a clear understanding of some aspects of Thomas Berry's proposal.  Overall I find that while many quotes are used, the article remains at a vague and at times superficial level.  More depth and secondary refs would assist.

Generalizations need to be taken out. For example, the second paragraph:  This paragraph is too general.  Who is the 'we'?  All these claims need to be more specific, less absolute, and contextualized, with references.

What must also be changed are the many generalized claims about 'world religions',  'we', spirituality, authenticity, etc.  

The article does not make an 'argument'.  It is a presentation of some of Berry's contributions.  The author needs to take the reader from one place to another.  A clear flow is needed for what is presented, why and where is this going.  These need to be added and made more clear.  

The conclusion needs to be redone as it is vague, generalized, and could be divided into two sections: implications - in a more developed manner,  and a conclusion.

The author needs to not just give an overview of the relevant aspects of Berry's work, but also the relevance and implications  - as it currently reads more as an exposé rather than an emphasis on the relevance.

Overall the articles needs to be tightened up, and strengthened with greater precision, fewer generalizations, more definitions, and a greater use of the many other secondary sources and academic publications on Berry's work.

Author Response

Thank you for your comments. I have revised the article and endeavoured to make a clearer argument for the ongoing relevance of Berry's thought to the renewal of catholicity. I have organized the article into subsections that help the reader move forward with the argument. The conclusion is more specific. I have included specific examples to replace the places where the article is vague or too general. I have included more of the secondary sources on Berry's work.

Reviewer 2 Report

Overall this article does not provide much that is not already very well know in the fields of environmental and ecological ethics, eco-theology or spirituality.  It might serve as an introductory piece for lay readers or undergrads. This would be best described as a review or a report. The argument for "catholicity" seems rather weak, perhaps because there is no clear distinction made between that and the more generally goals of inter-religious dialogue.  Catholicity needs to be more clearly distinguished defined. There is no discussion of the practical aspects of how Berry's forays into such inter-religious dialogue worked nor how today's practitioners could adjust to the current - arguably more urgent state of affairs.

Author Response

Thank you for your comments. I have revised the article and made a clearer argument for the ongoing and critical relevance of Berry's thought to a renewal of catholicity. I have made references to specific areas of catholic thought that need to be developed and how Berry's thought can contribute to that development. I have included some examples of how Berry's thought is influencing some concrete programs and places that are committed to the renewal of what 'catholic' means, most particularly in terms of developing an integral ecology.

Round 2

Reviewer 2 Report

Over all, there is significant improvement in this version of the article.  the remaining difficulty is the presumption that the reader will have a clear understanding of what the author means by "catholicity"  or "Catholicity."

Catholicity is a rather faddish, cliché, colloquial, niche term that is not precisely defined anywhere in this work. The readers who are familiar with the series edited by Ilia Delio might have some clue - but it is not clear that this author is consistent in utilizing Delio's definition - or?

What if any is the difference between Catholicity and catholicity? See lines, 89, 279, 286, 321, 366, 383, 463, 497, 550, 556, 562.

Author Response

I have clarified that I am using 'catholicity' in the way that Ilia Delio does for this issue. I have also indicated that this expanded sense of 'catholicity' needs to be attended to by the contemporary Catholic Church.